# Medical Aspects of mTOR Inhibition in Kidney Transplantation

**DOI:** 10.3390/ijms23147707

**Published:** 2022-07-12

**Authors:** Elena Cuadrado-Payán, Fritz Diekmann, David Cucchiari

**Affiliations:** 1Department of Nephrology and Kidney Transplantation, Hospital Clínic, 08036 Barcelona, Spain; ecuadrado@clinic.cat (E.C.-P.); fdiekmann@clinic.cat (F.D.); 2Laboratori Experimental de Nefrologia i Trasplantament (LENIT), Institut d’Investigacions Biomèdiques August Pi i Sunyer (IDIBAPS), 08036 Barcelona, Spain; 3Red de Investigación Renal (REDINREN), 28029 Madrid, Spain

**Keywords:** kidney, transplant, kidney transplant, immunosuppression, mTOR, mTOR inhibition

## Abstract

The advances in transplant immunosuppression have reduced substantially the incidence of kidney graft rejection. In recent years, the focus has moved from preventing rejection to preventing the long-term consequences of long-standing immunosuppression, including nephrotoxicity induced by calcineurin inhibitors (CNI), as well as infectious and neoplastic complications. Since the appearance in the late 1990s of mTOR inhibitors (mTORi), these unmet needs in immunosuppression management could be addressed thanks to their benefits (reduced rate of viral infections and cancer). However, management of side effects can be troublesome and hands-on experience is needed. Here, we review all the available information about them. Thanks to all the basic, translational and clinical research achieved in the last twenty years, we now use mTORi as de novo immunosuppression in association with CNI. Another possibility is represented by the conversion of either CNI or mycophenolate (MPA) to an mTORi later on after transplantation in low-risk kidney transplant recipients.

## 1. Introduction

The landscape of kidney transplantation has changed notably, moving from an incidence of acute kidney graft rejection of >80% in the early ages to <10% nowadays, as a result of the advances in transplant immunosuppression. Initially, it was based on steroids and azathioprine (AZA), but the current gold standard, also recommended by the KDIGO guidelines, includes first-line induction therapy with basiliximab in association with a calcineurin inhibitor (CNI) (preferably, tacrolimus, TAC) and an antiproliferative agent (preferably, mycophenolate), with the possibility of early steroid withdrawal in low-risk recipients. In recipients with a high immunological risk, the suggested first-line induction therapy is represented instead by lymphocyte-depleting agents [1]. Moreover, in recent years the focus has moved from preventing rejection to preventing the long-term consequences of long-standing immunosuppression. Among them, nephrotoxicity induced by CNI and infectious and neoplastic complications have to be highlighted.

Since the appearance in the late 1990s of mTOR inhibitors (mTORi), these unmet needs in immunosuppression management could be addressed [2,3]. It is generally accepted that mTORi reduce the incidence of viral infections in kidney transplant recipients, especially Cytomegalovirus (CMV) [4], and may reduce the incidence of neoplastic complications in the long term, even though the most convincing data are about non-melanoma skin cancers [5,6].

## 2. Pharmacology of mTOR Inhibitors

In 1964,on the South Pacific island of Rapa Nui (Easter Island), a Canadian expedition took soil samples, aiming to discover novel antimicrobial agents. Later, it was discovered that one of the compounds extracted by Streptomyces hygroscopicus had immunosuppressive, antitumor and antifungal activity [7,8,9]. These properties were due to the interaction of the molecule with an immunophillin (FKBP-12) that was necessary to inhibit cell growth and proliferation [10]. Curiously, the same immunophillin mediates signal transduction for TAC [11]. This substance was named Rapamycin (RAPA) on behalf of the name of the island and clinically is known as sirolimus (SRL).

In the following years, different groups discovered that the target of RAPA was a multiprotein complex analog to the yeast TOR gene [12,13,14,15], so it was named as mechanistic (formerly mammalian) Target of Rapamycin (mTOR) [15]. Further discoveries established that mTOR is a serine/threonine protein kinase that forms the catalytic subunit of the two largest multiproteic complexes, mTOR Complex 1 (mTORC1) and mTOR Complex 2 (mTORC2) (Figure 1) [16]. Recently, mEAK-7 (mTOR associated protein, eak-7 homolog) was identified as a positive activator of mTOR signaling via an alternative mTOR complex and it has been theorized that this novel complex is a third member of known mTOR complexes, mTORC3 [17,18].

The key components associated with mTOR in mTORC1 are RAPTOR (Regulatory Associated Protein of mTOR) and mLST8. In turn, two inhibitory components of the complex are PRAS40 (Proline-Rich Akt Substrate of 40 kDa) and DEPTOR (DEP domain-containing mTOR-interacting protein) [19,20]. The components participating in the mTORC2 complex include mLST8, DEPTOR and RICTOR (Raptor-Independent Companion of mTOR), with its related regulatory proteins, mSin1 and Protor 1/2 [21,22,23].

In contrast to mTORC1, mTORC2 is not affected by acute treatment with RAPA. However, chronic RAPA treatment inhibits mTORC2 signaling; this seems to be due to the incapacity of RAPA-bound mTOR to incorporate into the newly assembled mTORC2 complexes [24]. Initially, it was thought that mTOR complex was cytosolic; later on, it became clear that upon activation, mTORC1 localizes at the surface of lysosomes in a process that is mediated by cytoplasmatic nutrients, especially amino acids [25].

### 2.1. Upstream Regulation of mTOR

The first upstream positive regulator of mTOR is a GTPase named Rheb that is controlled by a heterodimer complex formed by TSC1, TSC2 and TBC17, called TSC (Tuberous Sclerosis Complex) [26]. TSC inhibits Rheb by serving as its GTPase-Activating Protein (GAP) [27]. When TSC is inhibited and Rheb–GTP accumulates, it is free to activate mTOR, even if this mechanism of activation is poorly understood [28]. TSC represents the cornerstone to which many pathways converge to activate mTOR through its inhibition, including insulin/Insulin-like Growth Factor 1 (IGF-1)/Akt pathway and the Ras/Erk pathway [29,30]. Other pathways that activate mTOR through TSC inhibition are Wnt and TNF-alpha via IKKβ [31,32] (Figure 2). These inhibitions are carried out by phosphorylation operated by the different kinases (Erk, Atk, IKKβ). Curiously, phosphorylation operated by other pathways activates TSC thus inhibiting mTOR activity. These include AMPK in response to low intracellular levels of energy (increased AMP/ATP ratio) and REDD1 in response to low oxygen tension [29,33]. Upon activation mediated by Rheb, mTORC1 localization to the lysosomes surface is promoted by another family of GTPase (Rag) that interacts with RAPTOR. Rag activation is me-diated by branched amino acids such as leucine and arginine [34,35,36,37].This means that mTOR activation by growth factors is enabled only in presence of nutrients. As a matter of fact, insulin is able to activate mTOR via Akt through TSC2 inhibition [29]. This makes sense as proliferation and cell growth promoted by mTOR is subjected to energy availability and puts mTOR at the crossroad of this major signaling pathway.

On the other hand, the only mechanism that seems to activate the mTORC2 complex is the insulin/PIK3 pathway. Curiously, mTORC2 is also regulated by mTORC1, as the latter inhibits the insulin/PIK through Grb10 as a negative feedback loop [38].

### 2.2. Downstream Activity of mTOR

Proliferation, cell growth and migration are energy-consuming, so it makes sense that mTOR activation by these processes is effective only in the presence of an anabolic state. As a matter of fact, the presence of branched amino acids is an essential step for mTOR activation, as stated above [35]. Moreover, low-energy states activate TSC, leading to mTOR inhibition; these include AMPK activation by a reduced ATP/AMP ratio and low oxygen tension [31,32]. In addition, DNA damage blocks mTOR, as p53 target genes, including AMPK, TSC2 or PTEN [39]. In the presence of a positive environment, mTOR is finally able to start its effector activities, which include protein synthesis and catabolism, and orchestrating lipids, nucleotides and glucose metabolism.

If we focus on immune cells, mTORC1 switches the metabolic phenotype of T cells from a catabolic to an anabolic state in response to proliferation stimuli. In this case, the main stimulus that drives proliferation of activated T cells is represented by IL-2 [40]. Specifically, when T cells are activated by an MHC-presented antigen in an inflammatory microenvironment, calcium channels open and activate calcineurin. This is a phosphatase that dephosphorylates a family of transcription factors called Nuclear Factor of Activated T-cells (NFAT), which enter the nucleus and activate the transcription of IL-2. The IL-2 Receptor (CD25) activated by autocrine-produced IL-2 drives the proliferation signal through PIK3 and Akt downstream. TSC inhibition by Akt finally leads to mTORC1 activation [41], which arranges the metabolic state and differentiation of activated T cells (Figure 3).

Initially, it was believed that RAPA could exert its immunosuppressive effects simply by inhibiting T cell proliferation. As a matter of fact, mTOR activation degrades the cell cycle inhibitor p27 and increases the expression of cyclin D3 [42]. Surprisingly, IL-2-mediated proliferation of T cells is only slightly affected by mTOR-selective deletion in mice models [43]. However, the mTOR-null CD4-T cells failed to differentiate into Th1, Th17 and Th2 under strong polarizing conditions. This has been deemed to decrease STAT4, STAT3 and STAT6 in response to the skewing cytokines IL-12, IL-6 and IL-4, respectively [44]. This reduction in STAT activation decreases the expression of the transcription factors T-bet, RORγt and GATA-3,which are essential to drive T cell differentiation. These studies are in line with previous mechanistic studies in which the use of RAPA was able to generate Foxp3 + Tregs [45]. Even in the presence of co-stimulation, RAPA promotes tolerance through the activation of the Foxp3 promoter [46].

Further experiments demonstrate that upon differential activation of mTORC1 or mTORC2, T cell differentiation is skewed. In mice lacking Rheb (thus without mTORC1 activity), the activation of mTORC2 leads to Th2 differentiation through IL-4-dependent STAT6 activation [47]. mTORC2 inhibits the Suppressor of Cytokine 5 (SOCS5) that negatively controls the STAT6-mediated activation of GATA-3, the transcription factor responsible for Th2 differentiation. On the other hand, mTORC2 inhibition through selective RICTOR deletion leads to Th1 and Th17 differentiation [47,48]. mTORC1 activation inhibits the Suppressor of Cytokine Signaling 3 (SOCS3),which in turn inhibits STAT4 and STAT3. IL-12- and IL-6-dependent activation of STAT4 and STAT3 finally leads to an increased expression of T-bet and RORγt, the transcription factors responsible for Th1 and Th17 commitment [43] (Figure 4). Fewer data are available on the role of mTOR in B cells. In different experiments it has been demonstrated that conditional deletion of mTOR in either early (CD79a) or late-stage (CD19) B cells ultimately leads to major defects/ blocks in B-cell differentiation, proliferation and survival, and ultimately results in the inability of mice to mount specific antibody responses to antigen [49,50].

Regarding antigen presentation, mTOR also has a role in dendritic cell maturation and the expression of cell membrane MHC molecules. Maturation from bone marrow-derived cells into dendritic cells in vitro is a process inhibited by RAPA [51]. Moreover, dendritic cells exposed to RAPA promote T cell tolerance, thanks to a decreased expression of costimulatory molecules and pro-inflammatory cytokines [52].

## 3. Use of mTOR Inhibitors in Graft-versus-Host Disease

Rapamycin and its analogs have been increasingly used to prevent graft-versus host disease (GVHD) after bone marrow transplantation (BMT). GVHD still represents the major complication after BMT, resulting in life-threatening complications for the recipient. It occurs when T cells in the transplant become activated by alloantigens and subsequently destroy recipient tissues [53,54].

Whilst promising response rates particularly for the treatment of chronic GVHD have been reported, the toxicity profile particularly in combination with CNIs remains limiting. Also, they have been used for GVHD prevention as it has been shown that RAPA treatment can induce the accumulation of regulatory T cells (Treg) in the skin of mice after bone marrow transplantation. In a recent study, Scheurer et al. found that RAPA treatment can increase the immunosuppressive potential of myeloid-derived suppressor cells (MDSCs) whilst maintaining the anti-tumor cytotoxicity of T cells (graft vs. tumor) without impairing the induction of Treg in a bone marrow transplantation mouse model. However, other in vitro studies and clinical findings demonstrated that the development of RAPA resistance typically occurs [53,54].

Thus, future use of mTOR inhibitors may rather favour prophylaxis than treatment of GVHD. Here, combinations without CNIs may offer promising prophylactic regimens with low toxicity rates [54]

## 4. Use of mTOR Inhibitors in Kidney Transplantation

The current state of the art with mTORi is the quest to discover the optimal immunosuppressive schedule that could guarantee kidney transplant recipients the lowest incidence of rejection and the best safety and long-term renal function. Thanks to all the basic, translational and clinical research achieved in the last twenty years, we now use mTORi as de novo immunosuppression in association with CNI at trough levels of 3–8 ng/mL. Another possibility is represented by the conversion of either CNI or mycophenolate (MPA) to an mTORi later on after transplantation. This can be beneficial in cases in which CNI- or MPA-related toxicity are evident, such as nephrotoxicity, tremor, leucopenia, diarrhea or CMV replication, which warrant a change in the immunosuppressive schedule. In these cases, late conversion can be carried out safely for most patients, especially from MPA to mTORi.

Moreover, different combinations of mTORi with the other immunosuppressive drugs have been investigated. Due to the narrow therapeutic index and the vast effects induced by mTORC1 and mTORC2 on human health and metabolism, management of side effects was challenging and hands-on experience was needed. Initially, it was not even clear that checking the trough level was necessary [55], as some trials focused only on the oral dose and not on therapeutic drug monitoring [56]. The general feeling about mTORi in the transplant community fluctuated from enthusiasm to disappointment, and vice-versa, given the brilliant discoveries and the frustrating failures. As a matter of fact, what we know about mTORi in kidney transplantation derives from the sum of pre-clinical and clinical data that have highlighted the strengths and the weaknesses of mTORi in this setting.

### 4.1. Early Clinical Trials

In 1996, the first study about the use of mTORi in kidney transplant recipients was published. In this phase-I trial, Murgia et al. analyzed the tolerability and side effects of different doses of sirolimus (SRL, 1 to 13 mg/m2 daily, divided in two doses) in 40 stable kidney transplant recipients treated with cyclosporine and steroids. The authors did not observe any difference in terms of renal function, liver function tests, cyclosporine levels and blood pressure. Side effects were thrombocytopenia (dose-related) and mild leucopenia (dose-unrelated), as well as an increase in total cholesterol level, while triglycerides were not affected [57]. In a phase-II trial, 83 patients were randomized to receive either cyclosporine (200–400 ng/mL for the first two months and 100–200 thereafter, *n* = 42) or SRL, (target trough levels 30 ng/mL for the first two months and 15 ng/mL thereafter, *n* = 41) without induction [2]. The two drugs were associated with AZA and steroids. Results were comparable in terms of acute rejection (41% for SRL and 38% for CsA), while a consistent improvement in renal function was noted in the mTORi group, along with less incidence of tremor and hypertension. However, the very high trough levels reached with SRL were associated with leuco thrombocytopenia, dyslipidemia and mTORi-associated pneumonia [2].

It started to become clear that the advantage to substitute CNI with mTORi along with an antiproliferative agent (AZA/MPA) was a better renal function and a lower incidence of CMV infection. However, these advantages were outweighed by the higher incidence of metabolic and hematological side effects and, possibly, early rejection. In an attempt to mitigate SRL side effects and to assure a low incidence of rejection, it seemed reasonable to combine it at an optimal therapeutic dose, along with little exposure to CsA. A phase-III double-blind multicenter trial published in the Lancet in 2000 validated the benefit of the CsA and SRL combination in comparison with CsA and AZA [58].

However, other players entered the field. Mycophenolate (MPA) was gradually substituting AZA as the antiproliferative agent of choice, given its better safety profile and the reduced rate of rejection [59]. Induction with anti-CD25 antibodies was increasingly recognized to lower rejection [60] and the reduced nephrotoxicity of TAC in comparison with cyclosporine was becoming widely accepted [61]. Therefore, the search for the best immunosuppressive combination was still far from being achieved. In the early 2000s, the big question was the possibility to avoid the use of CNI in kidney transplantation and mTORi entered into this race as the drug that could replace CNI in the future care of kidney transplant recipients.

### 4.2. mTOR Inhibitors as an Alternative to Calcineurin Inhibitors

The SPIESSER group confirmed the advantage of SRL over CsA in terms of renal function through the trial based on ATG induction. Patients were randomized to receive either SRL (*n* = 71) or CsA (*n* = 74) in combination with MPA and steroids (withdrawn at the end of month 5) [62]. SRL was started 2 days after transplant with a loading dose of 15 mg for 2 days, followed by 10 mg daily, and later adjusted to maintain trough levels of 10–15 ng/mL. Cyclosporine was started 48 h after transplantation as well, with target trough levels of 150–250 ng/mL for the first three months and 75–150 ng/mL thereafter. A non-significant increase in acute rejection was observed in the SRL group (14.3% versus 8.6%, *p* = 0.40) and there were no differences about patient and graft survival. A higher rate of study drug discontinuation occurred in the SRL group (28.2% versus 14.9%) and 12-month renal function of the on-treatment population at twelve months was better with SRL (68.7 ± 19 mL/min versus 60.1 ± 13.8 ng/mL).

On the other hand, if the combination of SRL/MPA was compared with TAC rather than CsA as the CNI of choice, the results were different. A meta-analysis published in 2006 made it clear the renal function benefit given by SRL took place when the associated CNI was CsA, but not TAC [63].

In summary, TAC was associated with the same results of SRL when combined with MPA and was better tolerated. At the beginning, the side effects associated with mTORi were accepted, especially for the advantage that SRL had in comparison to CsA in terms of improved renal function. However, when the comparator was no longer CsA, but the less-nephrotoxic TAC, the benefit in renal function disappeared and raised the reasonable doubt of whetherthe price to pay in terms of mTOR-associated side effects was worth the cost.

It was at this time when one of the most important trials in the recent history of transplant medicine appeared: the ELITE–Symphony [64]. It was a 4-arm study with 1645 kidney transplant recipients included, in which baseline immunosuppression was based on mycophenolate and prednisone, along with one of the following: (1) standard-dose CsA, (2) low-dose CsA, (3) low-dose TAC, (4) low-dose sirolimus (SRL). Groups 2, 3 and 4 also received induction with daclizumab. The combination that proved to be superior in terms of rejection, graft survival and renal function was undoubtedly the one based on the anti-CD25 antibody, low-dose TAC, mycophenolate and prednisone. SRL had far worse results in terms of BPAR (37.2%) compared to the standard-dose (25.8%) and low-dose (24.0%) CsA and low-dose TAC (12.3%).

One-year allograft survival was notably worse in the SRL group compared to the TAC group (91.7% versus 96.4%, *p* = 0.007). Only CMV infections were lower in the SRL group (*p* = 0.003). Withdrawal from the treatment was 48.9% in the SRL group compared to 20.0% in the TAC group. These results ratified the inferiority of the mTORi/MPA combination when used de novo in comparison with a modern schedule based on anti-CD25 induction, TAC and MPA. It has to be noted that mean trough levels approached 8 ng/mL during the course of the whole study. This means that almost half of patients were above the target range and dose-related effects could ensue easily.

Therefore, following the results of these crucial studies on the combination of SRL/MPA in kidney transplantation, some conclusions could be drawn. First, mTORi could constitute an alternative to CsA, but not to TAC, as a CNI-free and induction-based regimen. Second, in a CNI-free regimen, in order to maintain acceptable low rates of rejection, mTORi have to be used at very high trough levels. Last but not least, use of high trough levels of mTORi is associated with unacceptable side effects in the modern era of kidney transplantation.

In any case, the aforementioned observation of reduced CMV infection with mTORi and the preliminary findings about their beneficial role in tumor prevention [6,65] incited transplant physicians to continue the search for the right schedule in which mTORi could be implemented without causing harm.

### 4.3. mTOR Inhibitors Used in Order to Convert or Suspend Calcineurin Inhibitors

mTORi could be used as maintenance immunosuppression, allowing withdrawal or substitution of CNI after transplantation in a safe time period, ideally during the first year. In this way, CNI-associated nephrotoxicity is reduced without increasing the risk of rejection (as it is present only in the most delicate period after kidney transplant). In the following years, many trials examined this possibility.

Schena et al. analyzed the strategy of late conversion from CNI to SRL in the CONVERT trial. This study included 830 patients that, after kidney transplantation (from 6 to 120 months), were randomized to continue with baseline immunosuppression (CNI, MMF/AZA and steroids, *n* = 275) or to switch the CNI to SRL (*n* = 555). The primary endpoint at 12 months (improvement in renal function) was met in the on-treatment population and no difference in the incidence of BPAR was noted. A post-hoc analysis revealed that patients who experienced most benefit with conversion were those with a baseline GFR > 40 mL/min and a urinary protein to creatinine ratio ≤ 0.11 [66].

In a similar study, the ASCERTAIN, late conversion from CNI to everolimus (EVL, target trough levels 8–12 ng/mL) or CNI minimization with the introduction of EVL (target trough levels 3–8 ng/mL) was carried out 6 months after kidney transplantation. Twelve months after conversion there was no difference in renal function (the primary endpoint), with a higher rate of adverse events in the CNI minimization (16.7%) or conversion groups (28.3%) compared to the control group (4.1%). As in the previous trial, patients with greater baseline renal function (GFR > 50 mL/min) were the ones who experienced the best improvement compared to the control group. Early conversion (before 6 months after kidney transplantation) from CNI to an mTORi was examined in other studies, such as the ZEUS, the ELEVATE, the CONCEPT and the ORION trials [67,68,69,70]. Better renal function was observed in the ZEUS and in the CONCEPT, but not in the ELEVATE and in the ORION. Even though graft and patient survival were similar when compared to the control group, biopsy-proven rejection and de novo Donor-Specific Antibodies (DSAs) were more frequent in the conversion cohorts. Histologically, early conversion to mTORi is associated more frequently with signs of subclinical inflammation at kidney biopsy and this was identified as a predictor of poor long-term graft prognosis [71].

To conclude, converting CNI to an mTORi seems to be an acceptable option in low-risk kidney transplant recipients with good renal function and low proteinuria late after kidney transplantation. In the future, it would be interesting to analyze the long-term follow-up data of patients who were submitted to early or late conversion, looking for differences in hard outcomes.

### 4.4. Combination of mTOR Inhibitors with Calcineurin Inhibitors

All previous experiences agree that CNI are still necessary to maintain low rejection rates in kidney transplantation, so it makes sense to consider mTORi as an alternative to MPA in combination with CNIs and not as a substitute for them.

In 2015, based on early trials, a meta-analysis was published concluding that TAC/mTORi were associated with comparable results to TAC/MPA in terms of BPAR and patient survival, but results for renal function and graft survival were worse (death-censored graft loss with mTORi R.R. 1.31 [1.02–1.69]). Patients treated with mTORi had a higher risk of New-Onset Diabetes After Transplantation (NODAT), dyslipidemia, edema, lymphocele and thrombocytopenia, but a lower risk of CMV infection, malignancy and leucopenia [72].

It should be remarked that transplant physicians’ lack of experience with mTOR and the fact that in some trials CNI were not minimized could be responsible for these disappointing results in terms of renal function and graft survival. As a matter of fact, when EVL was used with optimal trough levels (3–8 ng/mL) in association with reduced-dose CNI, there was absolutely no difference in terms of hard outcomes and renal function [73].

Finally, this approach has been recently validated in the TRANSFORM trial, the largest trial ever performed in kidney transplantation. In this study, low-risk kidney transplant recipients were randomized to receive the mTORi Everolimus (EVL) or MPA in combination with a CNI (mostly, TAC, as CsA was planned to be included in fewer than 20% of cases) and prednisone [74]. Induction was center-based and in the mTORi group CNI minimization was planned with TAC target trough levels of 2–4 ng/mL and CsA levels of 25–50 ng/mL at 12 months. The use of EVL at an optimal and non-toxic trough level (3–8 ng/mL) proved to be non-inferior to mycophenolate for a binary composite endpoint at 12 months based on Biopsy-Proven Acute Rejection (BPAR) or Glomerular Filtration Rate (GFR) < 50 mL/min [74]. Graft and patient survival were not different, as well as the incidence of de novo DSAs. Higher discontinuation was noted in the EVL group (27.1 versus 18.7%) due to classical mTORi-associated side effects.

Reduced rates of infections were noted in the EVL group, especially CMV and BK. Two-year results of the trial were recently published, confirming the non-inferiority of the CNI/EVL regimen and also observing a reduced incidence of de novo DSA in the on-treatment population. Renal function was also not different between groups (52.6 versus 54.9 mL/min per 1.73 m^2^, respectively) [75]. When using SRL in place of EVL, such as in the RECORD trial, results were also comparable [76].

Another two recently published studies that deserve attention are the ATHENA and the US92 [77,78]. In the ATHENA study, patients were randomized to receive three immunosuppressive combinations: TAC/MPA (*n* = 205), TAC/EVL (*n* = 199) or CsA/EVL (*n* = 208), along with steroids and basiliximab induction. In contrast with the TRANSFORM, tacrolimus trough levels were not minimized in the EVL group (at 12 months, they reached an average of 6 ng/mL). The efficacy endpoint (a composite of BPAR, graft loss or death) was 13.0%, 24.6% and 9.8% for the TAC/EVL, CsA/EVL and TAC/MPA combinations, respectively (*p* = 0.260 for EVR/TAC and *p* < 0.001 for EVR/CsA versus MPA/TAC). Renal function endpoint revealed worse results for the two EVL groups in comparison with the MPA group (EVL/TAC 62.2 mL/min versus EVL/CsA 58.4 mL/min versus MPA/TAC 67.8 mL/min). While the results for CsA are not surprising, lower GFR in the EVL/TAC group was disappointing. A possible explanation given by the authors for this result is that TAC minimization was not planned in the EVL group. As a matter of fact, in the TRANSFORM trial there were no differences in 12-month renal function between groups. In a post-hoc analysis, the authors of the ATHENA study demonstrated that, indeed, EVL patients with low exposure to TAC (<5 ng/mL)did not have inferior renal function compared to the MPA group [77].

In the U292 trial, Qazi et al. essentially used the same approach of the TRANSFORM study in an American cohort of patients [78]. The primary endpoint, a composite of efficacy failure rate based on treated BPAR, graft loss, death or loss at follow-up, was missed by the EVL/TAC combination. This was due to the higher incidence of treated BPAR (19.1% versus 11.2%, *p* < 0.05), while graft loss (1.3% versus 3.9%; *p* < 0.05) and combined graft loss/death/loss at follow-up (6.1% versus 10.5%, *p* = 0.05) were significantly lower in the EVR/TAC group, with no differences in renal function. The authors explained this difference in treated BPAR with low exposure to EVL during the first two weeks after kidney transplant. Incidence of BPAR was indeed higher in those centers in which patients were under-exposed to the drug. Another difference with respect to the TRANSFORM trial was that high immunological risk was not an exclusion criterion [74,78]. Probably, in patients with high immunological risk, TAC should not be minimized early after transplantation.

The results of the last trials changed the perspective about the use of mTORi in kidney transplantation and the most recent meta-analysis confirmed that there are no differences for hard outcomes, including BPAR, patient and graft survival, in comparison with MPA [79]. These results differ from the previous meta-analysis published in 2015 [72] and are probably due to the better choice of mTORi trough levels and the careful minimization of CNI practiced in the TRANSFORM and the US92 trials.

## 5. Real-Life Use of mTOR Inhibitors in Renal Transplantation

All the lessons learned by all these randomized clinical trials taught the transplant community how to take advantage of the benefits of mTORi (reduced rate of viral infections and cancer), without paying an excessive price for their side effects. It is also important to bear in mind the strict inclusion criteria of the TRANSFORM trial. Patients at high immunological risk were discarded, as well as recipients of a Donors after Circulatory Death (DCD), which represent a valuable source of donors in many countries. In this field, a single-center propensity score analysis published in 2020 by our group [80] verified the real-life feasibility of using a TAC–mTORi combination scheme through 401 patients that were analyzed according to the baseline immunosuppression (TAC associated with either MPA or mTORi). mTORi were administered irrespective of the type of donor (non-heart beating or not) and the immunological risk of the recipient. Patients that would have not entered the TRANSFORM trial for these and other exclusion criteria accounted for 52.9% of the total population. Curiously, patients who met the TRANSFORM inclusion criteria (*n* = 186) had very similar results to that of the original trial, with no differences in terms of 1-year and last follow-up graft rejection and survival between the MPA and mTORi group. On the other hand, patients that could not have participated in the trial (*n* = 215), had better results for both outcomes. Another strong point in favor of mTORi was the evidence in all groups of better 1-year and last follow-up patient survival. A reduced rate of infection-related hospitalizations during the first year could partially justify this finding. On the other side, a higher incidence of drug discontinuation was observed in the mTORi group due to classical side effects, including hypercholesterolemia, proteinuria, surgical-associated complications, etc., as well as beneficial effects (reduced CMV reactivation and total number of infections requiring hospitalization).

A difference worthy to mention with respect to the TRANSFORM trial was the higher trough levels of TAC in patients treated with mTORi; this may also justify the decreased incidence of rejection in this group. This different attitude about TAC/mTORi trough levels was not associated with a worse 1-year renal function and higher chronicity scores at protocol renal biopsy [80].

In a sub-analysis of the same population focused on high immunological risk patients, defined as a baseline cPRA ≥ 50% (*n* = 71), the combination TAC + mTORi was associated with better results in terms of 1-year rejection-free survival compared to TAC + MPA (incidence of biopsy-proven acute rejection was 15.2% versus 36.8%, respectively) [81]. This striking difference in results in comparison with the US92 trial [78] is probably attributed to the higher TAC trough levels employed [80,81]. This probably indicatesthat in the high immunological risk population, TAC should not be minimized as in the low-risk population studied in the TRANSFORM trial.

## 6. Practical Use of mTOR Inhibitors in Kidney Transplantation—Troubleshooting

The two mTOR inhibitors commercially available and approved for use in kidney transplantation can be started soon after surgical intervention at a dose of 1–2 mg qd (Sirolimus, SRL) or 1–1.5 mg bid (Everolimus, EVL), with the aim to reach trough levels of 3–8 ng/mL. During the first weeks after kidney transplant, it is advisable, however, to maintain trough levels in the range of 3–5 ng/mL.

In our center, SRL and EVL are associated with TAC in order to reach a sum (TAC + mTORi) of trough levels of 8–12 ng/mL [80,81]. This sum can be reduced to 8–10 ng/mL at 6–12 months after kidney transplantation, according to individual assessment of rejection risk. Particularly, TAC can be minimized to <5 ng/mL after 6–12 months in the low-risk population according to the TRANSFORM experience [68]. In patients with high immunological risk, it is advisable not to minimize TAC during the first year after kidney transplantation and to consider reducing trough levels thereafter, according to local center policies and, preferably, to the results of per-indication or per-protocol kidney graft biopsies.

Induction should be based on individual risk assessment depending on the immunological risk (i.e., anti-CD25 antibodies for low-risk patients and anti-thymocyte globulins for the high-risk population).

Contraindications for the start of de novo mTOR inhibitors in kidney transplantation include: a previous history of intolerance or side effects with mTORi, chronic obstructive pulmonary disease, central obesity that could impair surgical wound healing, thrombotic microangiopathy as the cause of end-stage renal disease, and any patient at risk of surgical complications and possibly re-intervention. Patients that could benefit most from the use of mTOR inhibitors are those with a history of virally induced cancers and who are at risk of developing CMV disease or BK nephropathy.

Advantages for the use of mTOR inhibitors in comparison with MPA include, undoubtedly, less incidence of viral infections (especially, CMV and BK), less neutropenia and low blood platelets, and a possible reduction in long-term incidence of solid neoplasia, especially for non-melanoma skin cancer in which the evidence is more convincing [5,6]. Moreover, in low immunological risk patients, mTORi could allow safe minimization of CNI, which in the long term could theoretically prolong graft survival.

The most common side effects associated with the use of mTOR inhibitors are listed in Table 1, along with a list of possible solutions.

## 7. Conclusions

In conclusion, mTORi provide a valuable solution in today’s landscape of kidney transplantation as a first-line de novo therapy. They also represent a valuable option for replacing either TAC or MPA in stable kidney transplant recipients at low risk of rejection with immunosuppression-related side effects. Incidence of side effects has decreased in recent years, thanks to lower trough levels and hands-on experience. It should also be highlighted that, in most cases, side effects of mTORI can be easily managed, in the same way kidney transplant physicians are used to doing when managing the side effects of either TAC or MPA.

## Figures and Tables

**Figure 1 ijms-23-07707-f001:**
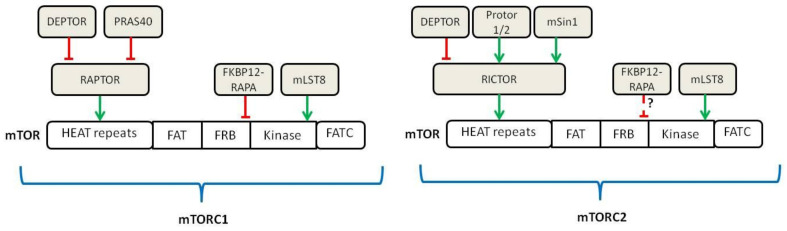
Schematic of the components belonging to mTORC1 and mTORC2 (adapted from [16]). Green lines show activating signals, red lines show inhibitory signals, dashed lines indicate that the exact mechanism is unknown.

**Figure 2 ijms-23-07707-f002:**
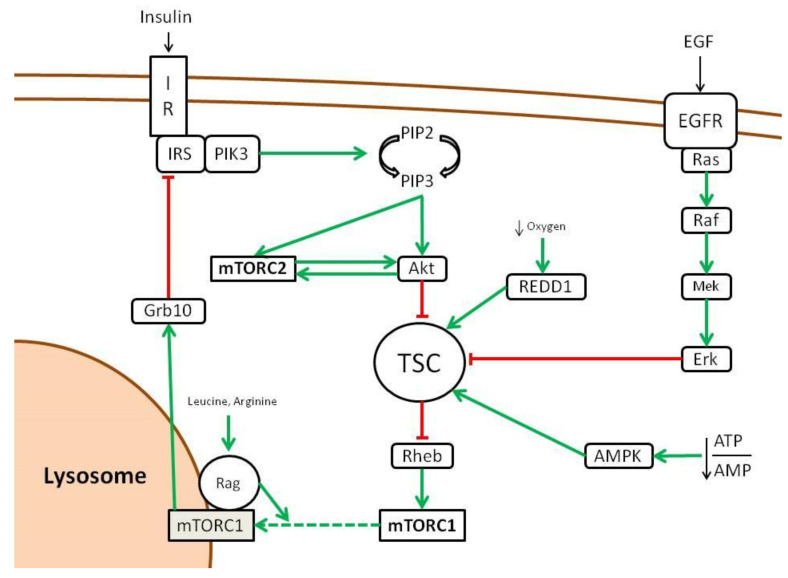
Upstream regulation of mTOR pathway (adapted from [16]). Green lines show activating signals, red lines show inhibitory signals, dashed lines indicate that the exact mechanism is unknown.

**Figure 3 ijms-23-07707-f003:**
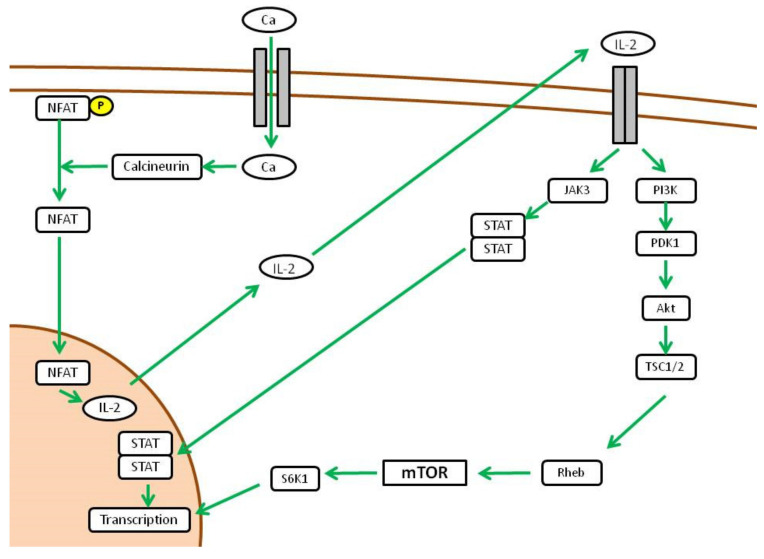
Downstream activity of mTOR pathway (adapted from [16]).

**Figure 4 ijms-23-07707-f004:**
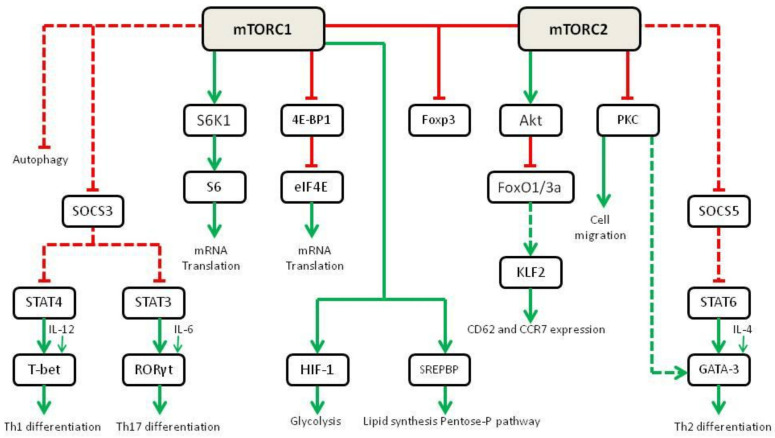
Role of mTOR in cell metabolism and differentiation in immune cells (adapted from [43]). Green lines show activating signals, red lines show inhibitory signals, dashed lines indicate that the exact mechanism is unknown.

**Table 1 ijms-23-07707-t001:** Most common side effects of mTOR inhibitors in kidney transplantation with a list of possible solutions.

Side Effect	Solution
Neumonitis	Discontinue mTORi.
Thrombotic microangiopathy	If clinically evident and in case of rejection, consider discontinuing mTORi. If it is only a finding in renal biopsy without clinical deterioration, consider reducing trough levels of either CNI or mTORi or both. In low-risk patients consider conversion from CNI to MPA.
Surgical scar infection or late healing	Switch to MPA until resolved and then switch back to mTORi.
Lymphocele	Switch to MPA until resolved and then switch back to mTORi.
Productive surgical drainage	Switch to MPA until resolved and then switch back to mTORi.
Post-transplant diabetes mellitus	Start of oral antidiabetic agent and/or insulin. Consider switching TAC to CsA.
Hypertriglicerydemia	Diet, weight loss, omega-3 fish oil.
Hypercolesterolemia	Diet, weight loss, statins, ezetimibe, fibrates.
Proteinuria	Consider using ACE inhibitors or Angiotensin Receptor Blockers.
Edemas	Consider using diuretics. In patients taking vasodilators (such as amlodipine), consider switching to another anti-hypertensive agent.

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
