# Peer review of "Medical Aspects of mTOR Inhibition in Kidney Transplantation"

_ijms, 2022, doi:10.3390/ijms23147707_

Round 1
Reviewer 1 Report
The authors, Cuadrado-Payán, Diekmann, and Cucchiari provide a comprehensive review on mTOR signaling, in an important context of transplant medicine and the unique challenge of long-term immunosuppression that patients face. The authors provide a wholistic background on the early days of kidney transplant medicine and the discovery of rapamycin. The authors provide the important discovery of the binding partner of rapamycin, mTOR. They also describe the numerous studies involved in mTOR signaling and the immune system. However, before the review is suitable for publication, several revisions should be addressed.
1) The authors should include a brief description of mTOR complex 3, which is defined by the binding partner mEAK-7 in mammalians. There are 3 manuscripts that should be summarized and cited: PMID: 29750193, 31288154, 33364499.
2) There are a handful of manuscripts describing the role of mTOR on B cells. Please expand on this topic, and at least, summarize and cite: PMID: 23858034, 34964999.
3) The authors should include a section on graft-versus-host disease as well.
Author Response
Thank you for the review and improvements proposed for our manuscript. We will now proceed to respond individually to each point.
1) The authors should include a brief description of mTOR complex 3, which is defined by the binding partner mEAK-7 in mammalians. There are 3 manuscripts that should be summarized and cited: PMID: 29750193, 31288154, 33364499.
Very interesting point. We have added that information to the manuscript
2) There are a handful of manuscripts describing the role of mTOR on B cells. Please expand on this topic, and at least, summarize and cite: PMID: 23858034, 34964999.
Absolutely right. Also, we have expanded that part with suggested bibliography
3) The authors should include a section on graft-versus-host disease as well.
Prior to the part of mTOR in kidney trasplantation, we have added a new section on graft-versus-host disease
Reviewer 2 Report
In this review, the authors provide a brief history of discovery of RAPA, its target TOR and summary of TOR complexes and up/downstreams. The major body of this review focuses on past evaluations of mTORi in transplantation especially kidney transplantation- with details on the outcome and analyses. Overall it is a well organized and well written review to provide a concise picture for years of work along the usage of mTORi in kidney transplantation, as well as observed side effects with potential solutions. There are some minor concerns that would need to be addressed:
Overall, the “mTORi” term is used across the manuscript. I assume this mTORi specifically refers to mTORC1 inhibitors (Rapalogs) but not dual mTORC1/mTORC2 inhibitors (like Torins). This would need to be clarified.
Conclusion section is quite short and can be further extended to include more perspectives and solutions for how to further improve mTORi in kidney transplantation, how to minimize risk factors to improve mTORi effects in antagonizing transplantation rejection, etc.
Page 2 line 61: ligands should be replaced with components- these are proteins.
Page 2 line 67: mTORC1 is localized on lysosome- this needs to be clarified. mTORC2 does not.
Section 2.1: This summary is pretty brief. mTORC2 is regulated by growth signaling as described, while mTORC1 is also activated by amino acids (through RAGs/RAGULATOR…) or cholesterol. This represents a critical mTORC1 regulatory axis that cannot be neglected.
Page 3 line 101: this sentence is not clear.
Author Response
Thank you for the review and improvements proposed for our manuscript. We will now proceed to respond individually to each point.
- Overall, the “mTORi” term is used across the manuscript. I assume this mTORi specifically refers to mTORC1 inhibitors (Rapalogs) but not dual mTORC1/mTORC2 inhibitors (like Torins). This would need to be clarified.
Thanks for the contribution. As you assume, mTORi are refered to mTORC1 inhibitors (Rapalogs). RAPA forms a complex with its intracellular receptor FK506-binding protein (FKBP12) to specifically inhibit mTORC1. RAPA effects on mTORC2 are variable, and require prolonged treatment.
- Conclusion section is quite short and can be further extended to include more perspectives and solutions for how to further improve mTORi in kidney transplantation, how to minimize risk factors to improve mTORi effects in antagonizing transplantation rejection, etc.
In the manuscript we have considered better to include the perspective and solutions for how to further improve mTORi and minimize risk factors in the section named “Practical use of mTOR inhibitors in kidney transplantation – Troubleshooting”. This is the reason for a quite short conclusion section. But if reviewers consider, we can move it.
- Page 2 line 61: ligands should be replaced with components- these are proteins.
Good point. We have replaced it
- Page 2 line 67: mTORC1 is localized on lysosome- this needs to be clarified. mTORC2 does not.
Thanks for such an important detail. We have clarified it.
- Section 2.1: This summary is pretty brief. mTORC2 is regulated by growth signaling as described, while mTORC1 is also activated by amino acids (through RAGs/RAGULATOR…) or cholesterol. This represents a critical mTORC1 regulatory axis that cannot be neglected.
Absolutely right. We have included this information in the manuscript
- Page 3 line 101: this sentence is not clear.
We have changed the order of the sentence to make it easier to understand.